# Older Adults’ Use of a Research-Based Web Platform for Social Interaction

**DOI:** 10.3390/healthcare11030408

**Published:** 2023-01-31

**Authors:** Annelie K. Gusdal, Ulrika Florin, Rose-Marie Johansson-Pajala, Caroline Eklund, Johanna Fritz, Petra von Heideken Wågert

**Affiliations:** 1School of Health, Care and Social Welfare, Mälardalen University, 631-05 Eskilstuna, Sweden; 2School of Innovation, Design and Engineering, Mälardalen University, 631-05 Eskilstuna, Sweden

**Keywords:** information and communication technology, loneliness, social isolation, social network, web platform

## Abstract

Loneliness and social isolation are triggers for unfavorable changes in older adults’ health and well-being. Information and communication technology (ICT) can be used by older adults to mitigate the negative effects of loneliness and social isolation. However, ICT needs to be customized to the specific needs and conditions of older adults. The aim of this study was to explore older adults’ use of a new, co-designed and research-based web platform for social interaction from the perspectives of older adults, researchers, and social services personnel. The study is an intervention study with a multimethod approach in which 20 older adults used the web platform for social interaction “the Fik@ room” for eight weeks. Quantitative and qualitative data were collected pretest, during the test, and posttest. The Fik@ room met the expectations of those older adults who completed the study. It enabled them to expand their social network and develop new friendships, but their experiences of loneliness were not reduced. The involvement of social services personnel in recruitment and support was important in facilitating older adults’ use of the Fik@ room. Our study contributes knowledge about a new, co-designed and research-based web platform, customized specifically for older adults, which is valuable in guiding the design and delivery of future web platforms for social interaction among older adults.

## 1. Introduction

With the older adult population expected to increase dramatically in the coming years, health issues involving loneliness and social isolation are growing concerns for health and social care services. Loneliness and social isolation are powerful triggers for a variety of disadvantageous changes in health and well-being [1,2,3,4]. Although loneliness and social isolation can occur at any stage in a person’s life, specific attention has been given to older adults because of higher rates of chronic illnesses and physical conditions, which may have an impact on loneliness and social isolation. Additionally, the increasing global trend of older adults living alone, and the fact that the size of older adults’ social networks and the extent of their social interactions decrease may impact loneliness and social isolation [5,6]. Loneliness and social isolation have also been exacerbated due to the COVID-19 pandemic with “stay-at-home” orders and recommendations of social distancing [7,8]. Overall, these conditions bring forth the importance of addressing loneliness and social isolation among older adults.

Digital technologies have become an integral part of modern societies [9]. One central trend has been the spread of information and communication technologies (ICT), which include devices such as smartphones, computers, laptops, and applications that provide access to information and electronic communications, e.g., sending text messages, engaging in video chats, and social networking [10]. ICT use may enable older adults to live independently for a prolonged time, and it can also play an important role in alleviating loneliness and social isolation [11,12,13,14,15]. It is noteworthy that loneliness (subjective distress due to an experienced discrepancy between one’s actual and desired social relationships) and social isolation (an objective and quantifiable lack of social connection or network) are often studied as separate entities, even though they synergistically interact with each other [16]. However, Newall and Menec [17] argue that although one can experience loneliness without social isolation and vice versa, exploring both subjective feelings and objective circumstances together provides a more holistic understanding of health and well-being, especially with respect to the outcomes of intervention studies.

One type of ICT used by older adults to mitigate the negative effects of loneliness and social isolation includes social networking services or other technology-based communication systems (e.g., Facebook, Instagram, Twitter, WhatsApp, E-mail, Skype), which focus specifically on connecting users with their family and/or friends and enhancing social relationships [11,12,15,18,19,20]. However, these existing social networking services mostly cater to the younger generations and do not consider the needs of older adults [11,15]. Furthermore, for online social networking services to be useful among older adults, the services must incorporate elements of supportive training and ongoing assistance to overcome barriers that have been found as specific to this population, such as privacy concerns, lack of experience, technology illiteracy, fear of using technology, lack of understanding of how to use the online interface, lack of family and friends to interact with on an online platform, and lack of user-friendly options designed specifically for an older population [13,21,22,23,24,25].

There is a growing interest in both the research community and in social services to design customized or new social networking services for reducing loneliness and social isolation among older adults. Our study contributes knowledge about older adults’ loneliness and social isolation/social networks when using a new research-based web platform for social interaction co-designed and developed with older adults and other important stakeholders. The aim of this study is to explore older adults’ use of a new co-designed research-based web platform for older adults’ social interaction from the perspectives of older adults, researchers, and social services personnel.

## 2. Materials and Methods

### 2.1. Design

The study includes an intervention study using a multimethod approach, combining qualitative and quantitative methods with the aim of gaining a more complete insight [26] into older adults’ use of a research-based web platform for social interaction. In addition, a participatory (user-involving) approach was a part of the study design, that acknowledges and implements users’ and stakeholders’ (older adults, researchers, and social services personnel) experiences throughout the development of the Fik@ room [27,28]. Quantitative and qualitative data were collected pretest, during the test, and posttest.

### 2.2. Setting, Participants and Recruitment

The study was conducted in the mid-east of Sweden in two different urban municipalities. One of the municipalities (Municipality A) had an ongoing digitalization project while the intervention study was conducted. This municipality was appointed by The Swedish Association of Local Authorities and Regions (SALAR) as a model municipality for the digital transformation of the care of older adults and received extra financial support. The participants included older adults and social services personnel. The social services personnel in both municipalities recruited older adults to participate in social interaction on the research-based web platform the Fik@ room by purposive sampling (see description in the next section). The recruitment was conducted through different channels: announcements in municipal meeting places for older adults, leaflets sent to households and followed-up with phone calls, local retirement organizations, advertisements in local newspapers and on Facebook, and through care personnel of older adults who received home care. Older adults were also “self-recruited” from adjacent municipalities, and older adults from the previous feasibility study [29] were offered participation as ambassadors/facilitators, but their experiences were not evaluated in the present study; hence, they were not research participants. The older adults without their own tablet either borrowed one from the municipality (Municipality A) or borrowed from the researchers (Municipality B) with the Fik@ room preinstalled. The inclusion criteria for the older adults were (a) age 60+, (b) experience of loneliness and wishes for increased social interaction/engagement, and (c) living within, or adjacent to, the geographical area of the two respective municipalities. The exclusion criteria for the older adults were (a) having moderate or severe cognitive impairment, and (b) not understanding or speaking the Swedish language.

In total, 66 older adults started participation in the study, and 20 completed the study. In total, 15 social services personnel recruited and supported the older adults during the intervention, together with home care personnel. Nine social services personnel participated in the posttest workshops. All social services personnel in both municipalities had older adults > 65 years as a focus in their work. Several of the social services personnel had previously participated in the development and testing of the web platform for social interaction [29].

### 2.3. The Research-Based Web Platform—The Fik@ Room

The Fik@ room is a web platform for older adults’ social interaction adapted for tablets, developed by researchers and co-designed with older adults, municipal social services personnel, and an IT company. The name the Fik@ room was suggested by older adults previously participating in workshops aimed at developing the web platform. The name was associated with the social conversations at coffee breaks (Fika in Swedish) at their former workplace. The functions of the Fik@ room are based on the older adults’ expressed needs and wishes in the workshops. The visual layout of the interface, with its functionalities, has self-explanatory navigation properties developed regarding contrasts, colors, gestures, and other information design specifics essential to the target group. The Fik@ room consisted of digital coffee/Fik@ tables, where limitless numbers of tables could be created, with seating for up to four persons per table. The conversations included either video, voice, or chat conversations about topics of the older adults’ own choice (see Figure 1). The Fik@ room also included a bulletin board where older adults could post messages and schedule meetings in the Fik@ room or at a physical location.

**Figure 1 healthcare-11-00408-f001:**
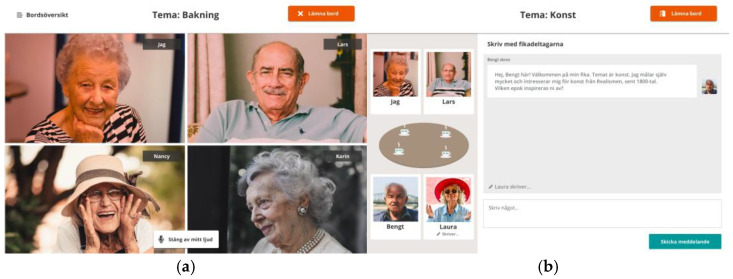
Print screen (**a**): A video conversation on the theme of baking (bakning in Swedish). Print screen (**b**): A chat conversation on the theme of art (konst in Swedish).

### 2.4. The Intervention

At the start of the 8-week study period running through May and June 2021, the older adults received welcome and information letters about the study and on how to download the application the Fik@ room (version number 0.0.82). The letter also included step-by-step instructions on how to get started and to use the Fik@ room (screenshots and descriptions of all functions and links to instructional videos), a statement of code of conduct when using the Fik@ room, and questionnaires to be completed and returned to the researchers before starting to use the Fik@ room (see Section 2.5). All older adults received unique usernames and passwords to the Fik@ room.

The older adults were encouraged to join conversations in the Fik@ room at the specified time of 1 p.m. on weekdays. These specified times aimed to increase the number of older adults being logged in at the same time. In addition to the specified times, the older adults were encouraged to use the Fik@ room as often and whenever they wanted. The bulletin board was vetted daily for any untoward messages breaking the code of conduct. During the 8-week study period, the older adults had access to support services provided by the municipality (A) or researchers (Municipality B) via telephone and e-mail.

### 2.5. Data Collection

Quantitative and qualitative data were collected pretest, during the test, and posttest. The older adults’ self-reported experiences of loneliness and social networks were collected pretest and posttest. Their patterns of use, and experiences of the support and information provided during the study, navigation and use of the Fik@ room’s functions were collected posttest. Researchers’ and social services personnel’s notes on older adults’ needs for support were collected during the study, and last, social services personnel’s experiences of older adults’ use of, and their incentives to participate in, the Fik@ room were collected posttest (see Table 1).

#### 2.5.1. Data Collected from Older Adults

The UCLA Loneliness Scale (Version 3) [30,31] evaluates self-reported loneliness and consists of 20 questions with scores ranging from 1 to 4 corresponding to *never*, *rarely*, *sometimes*, and *often*. Eleven of the questions are negatively worded, and nine of the questions are positively worded; thus, the latter questions’ scores were reversed. A higher total score indicates a higher degree of self-reported loneliness. Internal consistency was Cronbach’s alpha 0.92 pretest, and 0.90 posttest.

The Social network questionnaire evaluated the self-reported number of social activities and contacts and their satisfaction with them. The Social network questionnaire was constructed by the research team specifically for the Fik@ room project, inspired by Larsson et al. [32] and Rudman et al. [33], tested for face validity, and used in a previous feasibility study by Johansson-Pajala et al. [29]. It consists of 10 questions on the (a) frequency of social activities on and outside the internet with scores ranging from 4 to 1 corresponding to *every day*, *several days per week*, *a few days per week*, and *not at all*; (b) satisfaction with social activities on and outside the internet with scores ranging from 4 to 1 corresponding to *very satisfied*, *satisfied*, *dissatisfied*, and *very dissatisfied*; (c) the number of acquaintances, friends or relatives the older adult interacts with in various situations with scores ranging from 1 to 5 corresponding to *nobody*, *1–2 persons*, 3–*5 persons*, *6–10 persons*, and more than 10 persons; (d) the satisfaction with the number and quality of acquaintances, friends or relatives with scores ranging from 4 to 1 corresponding to *very satisfied*, *satisfied*, *dissatisfied*, and *very dissatisfied*; (e) overall satisfaction with existing social networks on and outside the internet with scores ranging from 4 to 1 corresponding to *very satisfied*, *satisfied*, *dissatisfied*, and *very dissatisfied*; and (f) overall change in the social networks on and outside the internet due to the pandemic with scores ranging from 0 to 10 on a visual analog scale (VAS) corresponding to *much deteriorated* and *much improved*. A higher total score indicates higher levels of satisfaction with one’s social networks. Internal consistency was Cronbach’s alpha 0.79 at both the pretest and posttest.

The Evaluation questionnaire evaluated the older adults’ self-reported experiences of their patterns of use, support and information provided during the study, and navigation and use of the Fik@ room’s functions. The evaluation questionnaire was constructed by the research team specifically for the Fik@ room project, tested for face validity, and used in a previous feasibility study by Johansson-Pajala et al. [29]. Of the 29 questionnaire items, 21 were quantitative statements on the (a) navigation and use of the various Fik@ room functions; (b) support and information provided; (c) the various Fik@ room functions addressing experienced loneliness and increasing social network with scores ranging from 5 to 1 corresponding to *highly agree*, *agree*, *neither agree nor disagree*, *disagree*, and *highly disagree*. There was also a sixth option of *No opinion/Do not know*, which received no score. A higher total score indicates a more satisfactory experience of the Fik@ room’s functions, support provided during the study, and the older adults’ experienced loneliness and increased social network in relation to their use of the Fik@ room. Internal consistency was Cronbach’s alpha 0.84 posttest.

The Evaluation questionnaire also included 8 items that were open-ended questions on the older adults’ patterns of use of the Fik@ room, their expectations of the Fik@ room, its importance for the older adult in relation to the on-going pandemic, older adults’ suggestions for improvement, which type of support the older adults preferred, and last, if they had any additional comments on the Fik@ room.

#### 2.5.2. Data Collected from Researchers and Social Services Personnel

Logbooks were kept by the researchers and social services personnel on older adults’ reasons for wanting to participate or not, their need for support, and which support was provided.

Workshops were conducted to explore the social services personnel’s experiences of the older adults’ use of, and incentives to participate in, the Fik@ room (present article) and determinants of implementing the Fik@ room in municipal settings (other article in press). Three workshops (two with Municipality A and one with Municipality B) were held with 3–6 social services personnel in each group in August and September 2021. For practical and pandemic-related reasons, all workshops were conducted digitally via an established video meeting tool. Each workshop lasted two hours and was digitally video recorded. Participation was voluntary, and all informants gave their verbal consent after receiving both verbal and written information. The workshops were moderated by two of the authors, with experience as moderators of workshops. The social services personnel both generated and analyzed data through a structured process that mixed individual and group activities, inspired by the effect modifier assessment (EMA) method [34]. The moderator balanced the workshops so that questions on the social services personnel’s experiences were evoked. The assistant moderator took observational notes and helped to recapture and summarize points of relevance at the end of each workshop. The participants then reflected, verified, and further developed the content, a recommended method of validation of workshops [35]. A summary based on observational notes and a debriefing session were performed by the moderator and the assistant moderator immediately after each workshop. The workshops were transcribed verbatim in their entirety.

### 2.6. Data Analysis

#### 2.6.1. Quantitative Data

Descriptive and inferential statistics of the quantitative data were calculated. The Wilcoxon signed-rank test was used to analyze the differences between the pretest and posttest. Missing values were 7.8% and at random, suggesting no bias, and multiple imputation of the item mean was used. The overall significance level was set at *p* ≤ 0.05. Statistical analyses were performed using IBM SPSS Statistics version 28.0 for Windows.

#### 2.6.2. Qualitative Data

Data from the evaluation questionnaire and logbooks on support were collated and analyzed using both an inductive and a deductive approach inspired by Neves et al. [36]. Data from the workshops were analyzed using a thematic analysis [37] with the QSR NVivo 1.5.2 [38] software program. Initially, all authors read the transcribed workshop material several times to obtain a general sense of the whole. Data extracts were identified, which generated a multitude of codes, which were then collated in a search for potential themes. Each step of the analysis was discussed by the authors at regular meetings until consensus was achieved.

## 3. Results

The results comprise the characteristics of the participants, experienced loneliness and social network, patterns of use of the Fik@ room, support and information, navigation and use of the functions, expectations of the Fik@ room, its importance in relation to the pandemic, and incentives for older adults to participate.

### 3.1. Participants

In total, 20 older adults completed the study, see characteristics in Table 2. There were several reasons for dropping out, such as illness, disinterest, or lack of energy and time.

Out of the nine social services personnel who participated in the workshops, six worked in Municipality A in managerial, digitalizational, care developing, and technical positions; three worked in Municipality B in managerial, senior guidance, and occupational therapist positions.

### 3.2. Older Adults’ Experienced Loneliness and Social Network

The UCLA loneliness scale [30,31] showed no significant difference between the pretest and posttest (Table 3). The four statements in the evaluation questionnaire that addressed loneliness and social networks versus the functions of the Fik@ room showed median values of 3 for all statements, but with the narrowest range and highest first quartile on whether the Fik@ room contains functions that are valuable in reducing the experience of loneliness. The social network questionnaire showed a significant difference in that older adults’ social networks increased due to the pandemic outside the internet between the pretest and posttest.

### 3.3. Older Adults’ Patterns of Use of the Fik@ Room

Most of the older adults used the Fik@ room 2–3 times/week or less than once/week (Table 4). The most common duration of each time in the Fik@ room was from less than 30 min to up to an hour, and video conversation was the most preferred type of conversation, as they preferred to see one another while talking, as these quotes exemplify: *Video facilitates social contact, it is easier to talk about your experiences* (older adult [OA]34); *It is video —nice to see each other, can show things, like flowers and such* (OA42).

To have specified times were important, as expressed by an older adult: *There needed to be some specified times as you cannot sit and log in all the time to see if anyone is there* (OA22). On the other hand, the fixed time of 1 p.m. on weekdays was not suitable for all, as it was either lunch time or that midday time split the day in two, as expressed in the following quote: *I’d like to change the time to approximately 10–11 a.m. and early evening approximately 6 p.m., which would be better than in the middle of the day* (OA9). Disappointment with the overall attendance was expressed: *It is sad that there seems to have been poor attendance. Many days you have not met anyone, usually the same few people inside* (OA13). Suggestions on how to improve poor attendance were to have more participants and, as previously mentioned, to have a morning time and an evening time instead of a midday time.

### 3.4. Support and Information, Navigationand Use of the Functions from the Perspectives of Older Adults, Researchers, and Social Services Personnel

Older adults’ experiences of support and information, and their navigation and use of the functions in the Fik@ room, i.e., connecting/creating conversations (video, voice, chat), creating/replying to messages on the bulletin board, and technicalities of sound, images, and keyboard, are summarized in Table 5.

Regarding support and information, the median values suggest that enough information was received prior to the start of using the Fik@ room and that personal support and written instructions were preferred over the instructional videos. Social services personnel in the workshops described that the written instructions were also supportive for them when informing the older adults about the study: *We need clarity and info sheets, I think it has been clear information, that it has been easy to convey* [to the older adults] *in various ways* (Municipality A, P4). The researchers’ and social services personnel’s notes on older adults’ needs of support during the 8-week study were mainly comprised of three difficulties: 1. login; 2. connecting to a table; and 3. the tablets’ image quality.

1. Login difficulties were the most common reasons for contacting the support team, and these difficulties were mainly due to the short validity period of the link sent to the older adults. Other login difficulties were due to older adults’ misprints of their e-mail address or password and were resolved through guidance while logging in. To simplify the log-in procedure, one member of social services personnel in the workshops suggested the following: *You should not have to write in the password and mail address, it could instead be a preprogramming where you just click on a link, then you enter the Fik@ room* (Municipality A, P3).

2. Connecting to table difficulties was the second most common reason for contacting the support team. The older adults were unable to connect to a table as the screen “froze”, or they simply did not understand how to connect. These difficulties were resolved through restarting the Fik@ room or through guidance while connecting to a table.

3. Image quality difficulties were the third most common reason for contacting the support team. The image was sometimes blurry or froze. This difficulty was resolved by system developers.

Regarding the navigation and use of the functions in the Fik@ room, the median values suggest that the older adults found it easy to connect/create all different types of conversations but not easy to create/reply to messages on the bulletin board. They did not find the image and sound quality, or the use of the keyboard, problematic. In older adults’ free-text answers in the Evaluation questionnaire on whether something needed to change or be developed in the Fik@ room, the following four quotes exemplify some of the older adults’ suggestions of improvement: *To have the opportunity to chat during ongoing conversations—as it can sometimes be useful to share written information* (OA20); *Want to be able to switch between the table and the bulletin board. Want to see how many who are logged in right now* (OA42); *There should first be a course for us who are not so good with technology* (OA16); *To have the possibility to connect to a “full” table of four persons. As the fifth to log in, you feel more alone when it is full on a table and no one else joins in to talk at the table that I created* (OA23).

It was also suggested to develop the Fik@ room through having a moderator as exemplified by the following two quotes: *A moderator/conversation leader who is involved and helps* (OA19); *To have a host/hostess who leads the conversation during certain times* (OA45). These quotes from older adults accord with the statements in the workshops with social services personnel who discussed the advantages of a moderator who could create new conversations/tables, coordinate, and get the conversation going and then leave. Preferably, this person could be a facilitator among the older adults as expressed by two social services personnel: *Maybe you should have someone there who talks to begin with, until people have gotten to know each other slightly more and feel comfortable and dare to talk, partly because there are other people you do not know and partly because it is a new forum* (Municipality A, P2); *What I think is needed is that someone is like a moderator, a conversation leader, it would be much easier, because then you might have the opportunity to create a relationship with someone who leads the conversation and who makes sure that everyone is involved* (Municipality B, P1).

### 3.5. Older Adults’ Expectations of the Fik@ Room and Its Importance in Relation to the Pandemic

In older adults’ free-text answers in the Evaluation questionnaire on whether the Fik@ room had met their expectations, as exemplified by the following three quotes: *Nice to meet new people during short but rewarding conversations about everything that just then felt important* (OA20); *My expectations have been met as I have seen and talked to people who are completely new to me* (OA3). Others’ experiences were less positive, as this quote exemplifies: *It did not feel like it was something for me. Did not feel that I wanted or could get a meaningful exchange socially in that way with complete strangers* (OA45).

The Fik@ room had been important for older adults’ social network in relation to the pandemic as expressed by an older adult: *Yes, it has been an opportunity to meet several people as the corona has been isolating. New personalities you got to ‘know’ and discuss with. I am satisfied* (OA35). For others, the Fik@ room had not been important as the social restrictions were lifted somewhat during the month of May: *Now in the late spring we could start spending slightly more time with others, even if you had to keep your distance* (OA3). The Fik@ room had most likely played a more important role if it had started earlier in 2021 instead of in late spring. Due to the warmer weather in the late spring of 2021, it was also easier to spend more time outdoors provided that a safe physical distance was kept. Others experienced the Fik@ room differently in relation to the pandemic as the following quotes exemplify: *Yes, it has been an opportunity to meet several people as the corona has been isolating. New personalities you got to ‘know’ and discuss with. I am satisfied* (OA35); *It is doubtful whether it had any major significance* [in relation to the pandemic] *but still nice to be involved and get food for thought about the importance of a social network and one’s wellbeing. Thank you*. (OA45).

### 3.6. Older Adults’ Incentives to Participate or Not Participate in the Fik@ Room—From the Perspective of the Social Services Personnel

Text written in bold italics represents the themes from the thematic analysis of the workshops.

In the workshops, social services personnel discussed older adults’ incentives to participate in the Fik@ room. If the older adults had ***adequate equipment*** and ***confidence and skills in digital technology***, the likelihood of their participation was described as greater. These older adults had experience of using the internet, e-mail systems and videocalls, and using their own equipment: *The more experience of technology you have, the better and more confident you become and thus more likely to participate* (Municipality A, P6). Similarly, if social services personnel had previously introduced the older adults to using different types of digital applications, reading the newspaper in an e-pub format, or playing digital games, they gradually accustomed themselves to the digital technology, which increased their incentives to participate in the Fik@ room.

Incentives for older adults to *not* participate were described as in part due to their ***lack of adequate equipment***. The Fik@ room was adapted for tablets and required Wi-Fi, and social services personnel described that some older adults had a desktop computer, or no computer at all, and no Wi-Fi. The older adults appreciated that they were offered a tablet to enable participation, but it did not alter some of the older adults’ disinterest in participating in the study. Other incentives for older adults to not participate were described as their ***lack of confidence and skills in digital technology***. The older adults’ digital experience was described as limited: *They can have a great mobile phone, but only use it to send messages and answer the phone* (Municipality B, P1). Social services personnel described that the older adults considered digital technology devices too complicated to use; either the simplest commands were hard to understand, the devices were hard to handle due to joint pain in the hands, or the touchscreens did not respond to the older adults’ dry fingertips. The social services personnel offered to teach and/or help the older adults and some appreciated the support, but it still did not change their mind about participating in the study. Other older adults were thought to be worried about not getting enough support and/or help due to the pandemic: *It is about feeling insecure about getting support, and usually we refer to the city library’s IT center for support, but with COVID they have closed this service, what they offer is online support, but that is what you need help with, so it becomes too complicated* (Municipality B, P1). Overall, the social services personnel described that older adults acknowledged their need to learn but considered themselves too old to learn new skills, irrespective of whether this involved creating an e-mail account or switching from their existing computer to a tablet. Social services personnel described that underlying older adults’ lack of confidence and skills was perhaps also a fear of technology, although it was not explicitly expressed by the older adults themselves as a reason for not participating in the Fik@ room: *I think of all movies, TV-series where someone accidentally presses a button and everything disappears, such conceptions still exist and can be difficult to get rid of* (Municipality A, P5). The social services personnel considered it important to understand the older adults’ fear but did not know how to address it: *Some have never been involved in technology, not even computers. It is a whole new world. Is it that you have heard about technology, that it can go wrong, why do you have that conception, when you haven’t even handled technology?* (Municipality A, P4).

***Lack of motivation and need*** were also described as incentives for older adults to not participate. Some did not see the point of being on the internet as it did not meet any of their needs: *Many chose not to join, so I think judging by that, there does not seem to be that need, and of all those who visit the municipal meeting places, they get their needs met there* (Municipality B, P2). It was also too complicated to grasp the “internet world” with its many risks of fraud: *There has been a lot of writing in the papers on scams and stuff so many were worried because of it, they simply did not dare because they did not know how to handle it* (Municipality A, P2). Social services personnel also discussed the dilemma of not reaching the appropriate target group. In the information material to the prospective participants, it was clearly stated that the goal of the study was to test a web platform to encourage social interaction among older adults. The words lonely or loneliness were not mentioned. Despite this positive approach, the social services personnel sensed that the older adults who did experience loneliness and needed social interaction did not wish to be seen as lonely and thus not wanting to participate: *It is slightly taboo to talk about loneliness, it is something you’re unlikely to be open about* (Municipality B, P3); this perhaps resulted in those older adults in most need of social interaction, unfortunately, missed out. Other older adults were described as not being interested at all in socializing via the internet and were content using their mobile phone and/or landline phone, or in no need for new social contacts at all. Other older adults were described as having skills in handling the internet through Instagram, Facebook, and Skype with good accessibility to friends and relatives on the internet and were not in need of social interaction with new people.

***Willingness to try something new*** was described as an important incentive for older adults to participate. In trying something new, older adults’ experienced feelings of joy and excitement. One social services personnel cited an older adult as follows: “*This sounds like fun. It’s a great idea. I would like to try this.*” (Municipality B, P3). Another described the willingness among older adults with the following quote: *At the meeting places for older adults where I have worked now for 11 years, we have maybe shown a movie clip from YouTube on the tablet, or shown other things such as games, cards, solitaire, a playfulness must transfer from the personnel and on to the older adults* (Municipality A, P3). Joy and excitement were described as closely entangled with the older adults’ curiosity: *There is a desire to try this, a curiosity, and we presented that you would meet more people in there, and you could talk about things* (Municipality A, P3). Social services personnel expressed several positive aftermaths from older adults’ participation in the Fik@ room: *Some have found each other and continued with our activities* [in the municipality], *we have cultural and digital activities, so that they meet there as well* (Municipality A, P3); *I know at least three ladies who after the Fik@ room ended they have continued to have contact with each other, and this is a win—win situation, it has truly given them something, it has meant new friends for them* (Municipality A, P4).

Of importance for the older adults’ incentives to begin participation in the study was described as whether ***personal contact*** had been established between the older adult and social services personnel prior to the study. The type of functions the social services personnel had in relation to the older adults was not relevant, it was rather the familiarity and trust previously created in the personal contact: *They say yes to participate in the Fik@ room, whether it was personnel who already had contact with them due to previous digital projects or the care staff, this intuitive feeling you can only have as a staff, I think, of what is right for this particular person* (Municipality A, P3). The social services personnel described how they knew what a particular older adult was most interested in and showed those things first in the Fik@ room: *Then we take the time to go back several times and do this together, step by step* (Municipality A, P5). It was also considered to be of importance for the older adults’ incentives to participate in the Fik@ room if the social services personnel themselves were knowledgeable of digital technology and fully invested and positive about the Fik@ room in their personal contacts with the older adults.

## 4. Discussion

Among the older adults who completed the study, the Fik@ room had met their expectations and it provided them with new contacts in a particularly difficult time due to the pandemic. Social interaction in the Fik@ room can be used to form new relationships and serve as a complement to older adults’ “off-line” relationships and activities. The Fik@ room was experienced as containing functions that are valuable for reducing older adults’ experienced loneliness and increasing their social network, albeit there were no significant differences in these two conditions between the pretest and posttest. Probable causes were the low attendance in the Fik@ room sessions due to a small sample and inconvenient specified times, the short duration of the intervention, and possibly a skewed recruitment of participants towards women (who experience less loneliness than men) and those not truly experiencing loneliness. Our intention was to target older adults in most need of social interaction and at high risk of loneliness. Risk factors for loneliness and social isolation include older adults who are not married/partnered, particularly those who have recently lost their partner; individuals with a limited social network and/or low levels of social activity; older adults with poor self-perceived health; and those with depression/depressed mood [39]. It is possible that the 46 older adults who dropped out of the study could have had some of the risk factors for loneliness, since lack of energy and their own or their partner’s illness were given as reasons for not continuing in the study. Furthermore, men’s loneliness and social isolation are significantly higher than women’s [40], particularly in an individualistic culture characteristic of Sweden, which is why it was a loss that men, who are generally in more need of social interaction, were not included in the study.

Adequate equipment, confidence and skills in digital technology, motivation and need, and last, willingness to try something new were incentives for older adults’ interest versus disinterest to participate in the study according to the social services personnel. The above incentives can be seen in light of how older adults still struggle with digital exclusion and technological barriers [15]. The near absence of social network platforms that meet the needs of older adults is identified as the major reason for older adults’ lack of involvement, despite the many potential benefits [25,41]. The lack of customized social network platforms may explain the positive turn back in recruitment of those older adults who had an established personal contact with the social services personnel who asked them to participate. In this contact, the uniqueness of the new, customized to older adults, Fik@ room could be explained and ease the older adults’ fear of getting involved.

In this study, the sample size was too small to generalize the quantitative results based on statistics pretest and posttest. Inconclusive findings from quantitative studies regarding loneliness and social isolation are frequently found in this research area and are mainly due to small sample studies, a lack of consensus on the definitions of the two terms loneliness and social isolation and the struggle related to the measurements of these outcomes, several methodological challenges, and confusion on how social network platforms or interactive devices are used [15,20,24,42]. Findings from qualitative studies in this research area provide more insight into participants’ experiences, which better guides future research on older adults’ use of social network platforms or interactive devices [24]. Thus, it is more productive to be guided by the present study’s qualitative findings than the quantitative findings.

The findings of the older adults coincide with social services personnel’s regarding older adults’ experienced support and information, and navigation and use of the functions of the Fik@ room. The Fik@ room was easy to use, but the tablet itself was sometimes difficult to handle, suggesting that the experience of the Fik@ room would improve if adapted for computers instead. As the Fik@ room was designed to be a highly secure web platform, difficulties with logging in was the primary concern, which should be facilitated in future studies. Furthermore, the findings from the older adults and the social services personnel coincide in their valuable suggestions on improvements of the Fik@ room for future use, e.g., to have a moderator at some of the conversation tables to facilitate the conversations and to include a chat function in the video conversations. Presumably, it is advantageous in an online social networking services intervention that the different users, in this study being the older adults and social services personnel, have a shared understanding of barriers and facilitators and that older adults are provided with support, as also highlighted in previous similar studies [21,43].

### Strengths and Limitations

Key strengths of this study are the multimethod approach in the data collection methods, as the qualitative data complemented and further explored the quantitative data and in relation to the groups of participants that resulted in the finding of a shared understanding of older adults’ and social services personnel’s experiences. The shared understanding can inform municipal strategists and managers in the development and implementation of the Fik@ room or some other web platform for social interaction. Furthermore, the Fik@ room builds on lessons learned in the previous feasibility study [29] in terms of how older adults were provided with support throughout the study period, and the Fik@ room was further customized to older adults’ needs regarding functions and visual layout.

Key limitations of this study pertain to the sample of older adults. Firstly, because of the small sample size, a generalization of the findings is not advised. Secondly, there was an overall heterogeneity in the sample, with respect to a majority of single households, which is of particular importance in a study exploring experiences of loneliness. Furthermore, there was a selection bias, as female participants were overrepresented among the older adults. Even though the terms “increase your social network/social interactions” were used by the social services personnel when recruiting the older adults, the presumptive participants were supposedly associated with loneliness and social isolation. As more women than men tend to report and disclose loneliness, it may be more acceptable for women than for men [44,45]; thus, the bias is not surprising. Additionally, the overrepresentation of women is consistent with the demographic distribution of older adults. Another limitation is that the older adults were not interviewed after their use of the Fik@ room; therefore, a deeper understanding of their experiences was not captured. The choice of using a tablet for the Fik@ room led to handling difficulties, and a better choice in future studies is to use stationary computers with which older adults presumably feel more comfortable and familiar, as computers have been utilized for a longer period.

## 5. Conclusions and Implications

The Fik@ room enabled older adults to expand their social network and develop new friendships with previously unknown people. Our study contributes new knowledge about a new, co-designed and research-based web platform, customized specifically for older adults, which is valuable in guiding the design and delivery of future web platforms for social interaction among older adults, especially in a municipal context with the involvement of social services personnel.

## Figures and Tables

**Table 1 healthcare-11-00408-t001:** Data collection at pretest, during test, and posttest.

Collected Data	Pre-Test	During Test	Post-Test	Data Collection Tools
Older adults’ experienced loneliness and social networks. Older adults’ patterns of use, experienced support and information, and navigation and use of the Fik@ room’s functions. Older adults’ expectations of the Fik@ room, its importance in relation to the pandemic, and suggestions for improvement. Researchers’ and social services personnel’s notes on older adults’ need of support. Social services personnel’s experiences of older adults’ use of, and incentives to participate in, the Fik@ room.	X	X	X X X X	UCLA Loneliness Scale Social network questionnaire Evaluation questionnaire Evaluation questionnaire Logbooks Workshops

**Table 2 healthcare-11-00408-t002:** Characteristics of the older adults (*n* = 20).

Variables	
**Older adults** (women/men)	20 (18/2)
**Age**, Md (range)	77 (63–93)
**Highest level of education**	
University/College, *n* (%)	9 (45)
Vocational school, *n* (%)	3 (15)
Secondary school, *n* (%)	2 (10)
Elementary school, *n* (%)	6 (30)
**Household**	
Living alone, *n* (%)	14 (70)
Cohabiting with partner, *n* (%)	5 (25)
Nursing home, *n* (%)	1 (5)
**Experience with digital technology**	
Computer/Laptop, *n* (%)	7 (35)
Tablet, *n* (%)	7 (35)
Smartphone, *n* (%)	14 (70)
Social media, *n* (%)	10 (50)
Digital assistant, *n* (%)	7 (35)

**Table 3 healthcare-11-00408-t003:** Experienced loneliness and social network at pretest and posttest ^a^ (*n* = 20).

Questionnaire and Variables (Min–Max)	Md (q1–q3) at Pretest	Md (q1–q3) at Posttest	*p* Value *
**UCLA Loneliness Scale** **(20–80)**	46.5 (32.5–52.25)	47.5 (37.25–51.75)	0.352
**Evaluation questionnaire**			
The Fik@ room contains functions that are valuable in reducing my experience of loneliness. (1–5)	NA	3 (2.25–4)	NA
The Fik@ room contains functions that are valuable in increasing my social network. (1–5) The bulletin board has an important function in the Fik@ room to reduce my experience of loneliness. (1–5)	NA NA	3 (1–4.75) 3 (1–4.75)	NA NA
The bulletin board has an important function in the Fik@ room to increase my social network. (1–5)	NA	3 (1–3.75)	NA
**Social network questionnaire** **(16–89)**	52 (47.75–55.50)	54 (47–59)	0.100
1. How often do you taken part in social activities with others *outside* the internet? (1–4)	2 (1–3)	2 (2–3)	0.446
2. How often do you take part in social activities with others *on* the internet? (1–4)	3 (1–3)	2 (1–3)	0.953
3. How satisfied are you with your social activities *outside* the internet? (1–4)	3 (2–3)	3 (2–3)	0.130
4. How satisfied are you with your social activities *on* the internet? (1–4)	3 (2–3)	3 (3–3)	0.672
5. How many acquaintances, friends, or relatives do you have that … (1–5) a. …you meet or speak with in an ordinary week? b. …can come to your home and feel comfortable at any time (even if you are in the middle of a meal, or if your home is untidy)? c. … have the same interests as you? d. … you can speak with openly without unease? e. … you can count on receiving support from if you get into trouble?	3 (2–3) 3 (2–3.75) 2 (2–3) 2 (2–3) 3 (3–3)	3 (2–4) 3 (2–3.75) 2 (2–3) 2 (2–3) 3 (2–3)	0.592 0.193 0.782 1.000 1.000
6. How satisfied are you (to what extent are your needs met) with the acquaintances, friends, or relatives that … (1–5) a. … you meet or speak with in an ordinary week? b. … can come to your home and feel comfortable at any time (even if you are in the middle of a meal, or if your home is untidy)? c. … have the same interests as you? d. … you can speak with openly without unease? e. … you can count on receiving support from if you get into trouble?	3 (3–3) 3 (3–3) 3 (2.25–3) **3 (3–3)** **3 (3–4)**	3 (3–3.75) 3 (3–3) 3 (3–3) **3 (3–4)** **3 (3–4)**	0.132 1.000 0.157 **0.020** **0.046**
7. How satisfied are you with your social networks outside the internet? (0–10)	3 (2.25–3)	3 (2–3)	0.593
8. How satisfied are you with your social networks on the internet? (0–10) 9. Have your social networks changed due to the pandemic outside the internet? (0–10) 10. Have your social networks changed due to the pandemic on the internet? (0–10)	3 (3–4) **2 (0–3)** 5 (3.5–7)	3 (3–3) **4 (2.25–5)** 6 (4.25–8)	0.632 **0.005** 0.528

UCLA Loneliness Scale—higher scores indicate a higher degree of loneliness. Evaluation questionnaire—higher scores indicate higher degrees of agreement with the respective variable. Social network questionnaire—higher scores indicate larger, improved, and more satisfaction with social networks. Md—Median (q1–q3)—the 25th and 75th percentiles, *n*—number, NA—Not applicable. ^a^ Wilcoxon signed-rank test, Asymp. Sig. (two-tailed) between pretest and posttest. * z values for the **significant** comparisons of scores are between −2.000 and −2.824, significant differences are written in **bold**.

**Table 4 healthcare-11-00408-t004:** Older adults’ patterns of use at posttest (*n* = 20).

Variables	*n*
**Frequency of use**	
2–3 times /week	8
Once/week	2
Less than once/week	9
Missing data	1
**Duration of time spent in the Fik@room on each occasion**	
More than 60 min	5
30–60 min	6
Less than 30 min	6
Missing data	3
**Preferred type of conversation**	
Video	11
Voice	0
Chat	1
No preferences	2
Missing data	6

**Table 5 healthcare-11-00408-t005:** Support and information, navigation and use of the functions at posttest (*n* = 20).

Variables—Evaluation Questionnaire (Min–Max)	Md (q1–q3) at Posttest
**Support and information**	
I received enough information to get started with the use of the Fik@ room. (1–5)	4 (3–5)
I needed to contact support to be able to log in to the Fik@ room for the first time. (1–5)	5 (1.25–5)
The written instructions on how to use the Fik@ room are a support for me. (1–5)	3 (1.25–4)
The instructional videos on how to use the Fik@ room are a support for me. (1–5)	2.5 (1–3.75)
I need to contact the support to be able to use the Fik@ room. (1–5)	2.5 (1.25–4)
**Navigation and use of the functions**	
It is easy to log in to the Fik@ room. (1–5)	4 (3–5)
It is easy to navigate in the Fik@ room. (1–5)	4 (3–5)
It is easy to “connect to conversations”. (1–5)	4 (3.25–5)
It is easy to “create a new conversation” (Video). (1–5)	4 (3–5)
It is easy to “create a new conversation” (Voice). (1–5)	4 (1.25–4.75)
It is easy to “create a new conversation” (Chat). (1–5)	4 (1–4)
It is easy to “create a new message” on the bulletin board. (1–5)	1 (1–4)
It is easy to reply to others’ messages on the bulletin board. (1–5)	1 (1–3.75)
I have a problem with the sound level/sound quality. (1–5)	2.5 (1–3)
I have a problem with the image quality. (1–5)	2 (1–2.75)
I have a problem using the keyboard. (1–5)	1 (1–3)
In the conversations, everyone has a friendly and respectful tone towards each other. (1–5)	5 (3.25–5)

Evaluation questionnaire—higher scores indicate higher degrees of agreement with the respective variable. Md—Median, (q1–q3)—the 25th and 75th percentiles, *n*—number.

## Data Availability

Data underlying the findings reported in the manuscript are available from the Qualitative Data Repository (https://doi.org/10.5064/F6PKM9XP accessed on 30 December 2022).

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
