# Peer review of "Older Adults’ Use of a Research-Based Web Platform for Social Interaction"

_healthcare, 2023, doi:10.3390/healthcare11030408_

Round 1

Reviewer 1 Report (Previous Reviewer 2)

Authors have made a number of appropriate modifications to qualify the results they present and to improve clarity of the instruments used in the study. 

1.  Lines 90-91 requires some editing for grammar “that acknowledges and implementS users’ and stakeholders’” making the nouns plural possessive.

Author Response

Authors’ Reply:
Thank you for your encouragement!

The two nouns are now in plural and thus congruent with the verbs (highlighted in yellow in 2.1 Design).

Reviewer 2 Report (Previous Reviewer 5)

In fact, I rejected the manuscript because I didn't see any major improvements in it, the manuscript norms also do not match the journal norms.

I can´t recommend acceptance of this manuscript.

Author Response

Authors’ Reply:
We are sorry to hear that you previously rejected our manuscript and that you do not see any major improvements. We do not know which of the reviewers you were in the previous round of submission and thus do not know whether you made any suggestions for improvement or not. As far as we understand, we have revised our manuscript according to reviewers’ suggestions in the previous round, with a few exceptions where we have explained why.

We are not sure what you mean by journal norms, our research described in the manuscript is within the journal’s aim and scope, the manuscript is formatted into the journal template and conforms to the reference system used by the journal.

Reviewer 3 Report (Previous Reviewer 1)

All suggestions and comments proposed have been taken into account.

Author Response

Authors’ Reply:
Thank you for your encouragement!

Reviewer 4 Report (Previous Reviewer 4)

Review for Manuscript entitled “OLDER ADULTS’ USE OF A RESEARCH-BASED WEB PLATFORM FOR SOCIAL INTERACTION”.

Thank you for providing me with the opportunity to review the manuscript entitled “Older adults’ use of a research-based web platform for social interaction”.

Congratulations for the great work done.

You can see that you have made the modifications proposed by all the reviewers. It is to be appreciated that all the requested changes have been carried out.

BACKGROUND

Introduction is written clearly.  

MATERIALS AND METHODS: 

The inclusion criteria appear in the publication but not the exclusion criteria.

It would be necessary to add criteria such as: no cognitive impairment, no depression...

Results: 

It is written clearly.  

DISCUSSION:

Limitations:

It is necessary to specify that there was a great variety in the sample with respect to the household. It is important to note that the number of people living alone in the home was greater than those living with others. This is a major limitation of the study, since the study is about loneliness. The heterogeneity of the sample is another limitation.

Conclusion: 

It is written clearly.  

Author Response

Reviewer 5 Report (New Reviewer)

I have reviewed the manuscript entitled: Older adults’ use of a research-based web platform for social interaction, the authors; it is a relevant work, with important and useful information for  older adults. However, it contains several issues, which are indicated below.

Line 14. The aim of the study…

Line 17. Briefly describe the platform “the Fik@ room”.

Line 19. It is important to properly define the results, including the main findings in statistical terms.

Line 25, 26. I recommend reducing the keywords to 5. Emphasizing the most relevant elements of the study.

The references are not appropriately written, according to the journal's criteria, you need reestructured them.

Line 81. The aim of this study...

 Point 2.1

· I recommend that in this part, it be defined that the informed consent of the participants was obtained and that the IBR approved the research protocol.

· The type of study carried out is mixed. There are no intervention studies, interventions are strategies that seek to identify findings from the manipulation of variables (10.3238/arztebl.2009.0262).

The authors need to describe more appropriately the characteristics of the studies carried out (i.e. cross-sectional, clinical, cohort?)

Lines 113-116. It is missing to define the exclusion criteria and the type of sampling.

The authors repeat information related to sample size limitations (lines 520 and 557), Is it recommendable delete one of them.

Line 153. Review final script.

Table 2. review and correct the value of n and the percentages in Household as there are errors.

ICTs can also cause isolation, loneliness and depression. (Consider briefly including their adverse effects in the discussion).

Round 2

Reviewer 2 Report (Previous Reviewer 5)

 I agree that this study will be published. 

Reviewer 4 Report (Previous Reviewer 4)

Review for Manuscript entitled “OLDER ADULTS’ USE OF A RESEARCH-BASED WEB PLATFORM FOR SOCIAL INTERACTION”. 

Thank you for providing me with the opportunity to review the manuscript entitled “Older adults’ use of a research-based web platform for social interaction”. 

Congratulations for the great work done.  

You can see that you have made the modifications proposed by all the reviewers. It is to be appreciated that all the requested changes have been carried out. 

After all the suggestions and changes made, the following publication is accepted for publication.

This manuscript is a resubmission of an earlier submission. The following is a list of the peer review reports and author responses from that submission.

Round 1

Reviewer 1 Report

The theme of the manuscript is quite interesting and is about a group that should not be abandoned. Older adults have a lot to contribute to our society and everything we can do to improve their well-being is good. At these ages, social relationships are undermined for various reasons and working on social interaction online with them is important. In order to increase the scientific quality of this article it would be advisable:

1. Include a hypothesis with the possible results that you expect to find in the main objective.

2. Igular format during the development of the text and tables.

3. Developing the discussion based on the order of the results found will facilitate their understanding.

Good work.

Reviewer 2 Report

Healthcare – use of research-based web platform for social interaction

Thank you for inviting me to review the current paper. Supporting social connection for older persons is an appropriate goal that technology may help. Numbered comments intended to help the authors improve the manuscript follow.

1.     Given the widespread recent attention paid to isolation and loneliness of older persons and the potential for web-based interventions to promote social connections among all age groups, I expect there is more current research the authors could draw on in the current paper.

2.     Some of the language at different points in the paper stereotypes and “otherizes” older adults, treating them as a uniform group, such as on p. 2, 46-49. People of any age can have chronic conditions and chronic conditions alone do not cause isolation or loneliness. Authors might qualify the statement by explaining that older adults have higher rates of chronic conditions and explaining how these can impact isolation and loneliness.

a.     Line 81 – barriers are not likely specific to older adults. Please qualify this statement.

b.     This diminishment of older adults as needing special help utilizing technology does not align with the results presented in table 1 indicating that half to most of the respondents at least are using smart phones and social media.

3.     Another main challenge is the very small sample that included people who were not particularly lonely. It is appears that authors did not have data from those who dropped out of the study; this would have been most valuable. Authors assert that the Fikk@ room met older adults’ expectations, but the high rate of attrition seems to suggest otherwise.

4.     Finally, qualitative data are limited to staff members’ perspectives and cannot be triangulated with older adults’ report, leaving this reviewer wondering how reliable staff were in their responses or if their answers might have been biased by their own stereotypes of older adults.

a.     For example, how can authors assert (lines 515-517) that personal contact was important to incentivize participation unless this information was known about those who persisted through the study and those who dropped out?

5.     This paper presents an interesting idea and incorporates information about developing and implementing the program with help from staff who are already connected to older adult clients. It may be suited to a pre-implementation paper.

a.     E.g., van Leeuwen and colleagues’ 2018 paper: https://journals.plos.org/plosone/article?id=10.1371/journal.pone.0208797

6.     Line 55, edit to read “modern societies, INCLUDING older adults…”

7.     Line 78 – please provide a citation for the statement that networking services do not consider older clients.

8.     Line 106 – I don’t believe “digitilisation” is the word authors mean to use as this involves converting analog information to 0s and 1s.

9.     Line 138 – “identified” may be a better word choice than formulated.

10.  Figure 1  - it is hard to tell if the caption refers to figures on the top and bottom or left and right. Perhaps it is simply the appearance on my page. Check that the language reflects the layout of the images as they will appear in print.

11.  Please provide the alphas for the measures listed pp. 5-6.

12.  Please clarify if the evaluation questionnaire was developed specifically for the Fik@ project. If so, explain how it was developed.

13.  Tables – check APA guidelines for formatting, please

14.  In sections of the discussion e.g., 409-411, it is difficult to tell if the authors are presenting an interpretation of results or quotes from a staff member. Much of this section of the paper seems to be a string of quotes.

Reviewer 3 Report

Introduction - the loneliness experience, as well as the various issues related to older adults, are hardly covered.

Methods - Why is font size different in some parts? Were the larger font parts been copied from another source?

Participants - very small number (20)

The letter that was sent to recruit participants needs to be attached.

The intervention is described vaugly

Measures - no psychometric data was provided for any of the measures used.

Reviewer 4 Report

Review for Manuscript entitled “OLDER ADULTS’ USE OF A RESEARCH-BASED WEB PLATFORM FOR SOCIAL INTERACTION”.

Thank you for providing me with the opportunity to review the manuscript entitled “Older adults’ use of a research-based web platform for social interaction”.

BACKGROUND

Introduction is written clearly.  

METHODS: 

The study is a mixed method. Consider modifying this in the text. 

“Mixed methods research combines elements of quantitative research and qualitative research in order to answer your research question. Mixed methods can help you gain a more complete picture than a standalone quantitative or qualitative study, as it integrates benefits of both methods.

Mixed methods research is often used in the behavioral, health, and social sciences, especially in multidisciplinary settings and complex situational or societal research.”

In this study, have you followed the Best Practices for Mixed Methods Research in the Health Sciences of NIH or the guidelines provided in Good Reporting of a Mixed Methods Study? This have to be explained and modified. 

Results: 

It is written clearly.  

Conclusion: 

It is written clearly.  

Reviewer 5 Report

Healthcare-2097998

I appreciate the opportunity of reviewing the manuscript entitled “Older adults’ use of a research-based web platform for social interaction”

The authors did not use the MDPI (Healthcare) Journal template, size, colours and font size do not comply with the journal's rules.

Keywords: Usually 3 to 5 keywords are used. The manuscript contains at least 11 keywords.

All sections need to be largely reworked; I write you here some of them.

Citations throughout the manuscript are not in accordance with the norms (MDPI).

The paper did not contextualize the main literature on the topic and the research question is not clear.

Introduction

I am not sure if the conceptual framework that allowed authors to perform this investigation is well described

Methods

Characterization of participants Sample characterization with (M±SD)

Results

In total, 20 older adults completed the study.

The course of the study should be described in detail. Why are only 20 of the 66 qualified people? What were the eligibility criteria?

Discussion

I would like to understand the new contribution of your manuscript to the science clarify in the discussion or conclusion section.

If I can see correctly, you use different colour in the text.

Conclusion

I consider the results the stronger section of the manuscript.
